# Accumulation of STR-Loci Aberrations in Subclones of Jurkat Cell Line as a Model of Tumor Clonal Evolution

**DOI:** 10.3390/genes14030571

**Published:** 2023-02-24

**Authors:** Natalya Risinskaya, Olga Glinshchikova, Tatiana Makarik, Yana Kozhevnikova, Julia Chabaeva, Sergey Kulikov

**Affiliations:** 1National Research Center for Hematology, 125167 Moscow, Russia; 2School of Medicine, Lomonosov Moscow State University, 119991 Moscow, Russia

**Keywords:** tumor clonal evolution, STR profiling, Jurkat subclones, WIL2-S subclones, loss of heterozygosity (LOH), elevated microsatellite alteration at selected tetranucleotide repeats (EMAST)

## Abstract

Many genetic markers are known to distinguish tumor cells from normal. Genetic lesions found at disease onset often belong to a predominant tumor clone, and further observation makes it possible to assess the fate of this clone during therapy. However, minor clones escape monitoring and become unidentified, leading to relapses. Here we report the results of in vitro study of clonal evolution in cultured tumor cell line (Jurkat) compared to the cell line of non-tumor origin (WIL2-S). Cell lines were cultured and cloned by limiting dilutions. Subclones were tested by short tandem repeats (STR) profiling. Spontaneous STR aberrations in cells of non-tumor origin occur in less than 1 of 100 cultured cells. While in the cells of tumor origin, new aberrations appear in 1 or even more of 3 cultured cells. At the same time, a significant relationship was found between the accumulation of aberrations in the pool of subclones and the rate of cell growth. One can speculate that this approach could be applied for the analysis of primary patient tumor cell culture to obtain information concerning the evolutionary potential of the tumor cells that may be useful for the selection of a therapy approach.

## 1. Introduction

Many genetic markers are known to distinguish tumor cells from normal. These are aberrations affecting extended chromosomal regions or the entire chromosomes: deletions, duplications, copy neutral loss of heterozygosity, various translocations, and lesions in short DNA fragments: point mutations, small deletions, insertions, etc., [1,2]. Some of these mutations are driver ones that impact the therapy choice and the outcome of the disease, some may simply serve as markers of a tumor clone [3]. In hematological malignancies it is possible to detect these aberrations at the onset, then during therapy. These markers are monitored to assess tumor clearance, minimal residual disease, and confirm relapse [4,5]. However, lesions found in the onset of the disease are often belonged to a predominant tumor clone, and further observation makes it possible to assess the fate of this clone during therapy. It is known that hematological malignancies, similarly to solid tumors, are characterized by tumor heterogeneity [6]. However, minor clones, not detected at the onset, escape monitoring and become unidentified leading to relapses. Moreover, time point analysis is unable to provide information concerning clonal evolution of tumor cells, that could explain the resistance of the tumor to chemotherapy and targeted therapy [7,8,9,10]. Here we report an in vitro model of the “natural” clonal evolution of T-lymphoblasts (Jurkat cell line). This cell line was established in the mid-1970s from the peripheral blood of a 14-year-old boy with T-cell leukemia. Now there are a lot of information about the change in the genetic landscape of the tumor from diagnosis to relapse in pediatric acute lymphoblastic leukemia and about mutational dynamics of early and late relapsed childhood all [11,12]. Therefore, we expected to see the emergence of new aberrant clones and changes in the mutational landscape during cell culture propagation. Subclones were obtained by limiting dilutions. The WIL2-S line of non-tumor origin was chosen as a control. The accumulation of aberrations was investigated by comparing the STR profiles of subclones. This robust research tool provides information on a variety of changes in the genome, including chromosome aberrations and small deletions and insertions of several nucleotides. Previously, we studied the STR profiles of tumor cells in patients with acute lymphoblastic leukemia, B-cell lymphomas, and multiple myeloma and observed two types of aberrations—loss of heterozygosity (LOH) and elevated microsatellite alteration at selected tetranucleotide repeats (EMAST) [13,14,15,16]. When verified by the microarray, the STR loci with LOH were always in the region of large deletions or a copy neutral loss of heterozygosity. EMASTs are short deletions or insertions in repeating tetranucleotide sequences that suggest a deficiency in the DNA repair complex. Sometimes new aberrations were added to the genetic profile in the recurrence of ALL. In multiple myeloma with plasmacytoma, differences in DNA profiles from plasmacytoma and bone marrow CD138+ cells were also observed [15,16]. Currently, many works are devoted to chemotherapy-induced clonal evolution. However, it is not possible in the clinical studies to assess in vivo the contribution of the initial evolutionary potential of the tumor to the emergence of new resistant clones during therapy. We propose an in vitro model to estimate the evolutionary potential of tumor cell clones.

## 2. Materials and Methods

### 2.1. Cell Cultures Cloning by Limiting Dilutions

Cell lines were obtained from ATCC. Jurkat (ATCC TIB152) and WIL2-S (ATCC CRL8885) cell lines were cultured in RPMI1640 medium (Gibco11875093) supplemented with 10% fetal bovine serum (FBS) (Gibco16140063). The cultures were diluted on the third day by removing part of the cell suspension and adding fresh medium. The inoculate concentration was 1 × 10^5^ cells/mL. The maximum cell concentration before seeding did not exceed 1 × 10^6^ cells/mL. Cell viability was assessed by staining with trypan blue (Gibco 15250061). Cell cultures with a viability of at least 90% were used for cloning. Cell lines were cloned by limiting dilutions at the rate of 1 cell per 3 wells of a 96-well plate. Cloning was performed in RPMI1640 medium supplemented with 20% FBS at 100 µL/well. A week later, the growth of the culture in the wells of the plate was monitored under a microscope, and 100 μL of the medium was added to the wells containing growing clones. At this stage, the wells contain from 10 to 100 cells. A week later, cells grown to the state of a monolayer were transferred into centrifuge tubes, leaving about 1/10 of the volume of the cell suspension in the wells. About 100 µL of fresh medium was added to these wells. Grown cultures were transferred to centrifuge tubes and sedimented for 5 min at 200 g.

### 2.2. DNA Isolation

DNA was isolated from the cell pellet using proteinase K digestion. The cell pellets were suspended in 20 µL of lysis buffer (0.5% NP40, 50 mM Tris (pH 8.3), 75 mM KCl, 20 µg of proteinase K), incubated at 56 °C for 1 h, followed by inactivation at 100 °C for 15 min. The lysates were centrifuged and 2 µL of the supernatant (1/10 of the total amount) was used for each PCR reaction [17]. Eighteen primary clones of the Jurkat line were analyzed, two of them were selected for further cultivation (B4 with stable STR-profile compared with STR-profile of Jurkat cell line and O1 with Y loss and 1 D12S391 LOH). After growth cells were re-cloned by the limiting dilutions, 180 B4 subclones and 381 O1 subclones were tested by STR-PCR. Similarly, 21 primary clones of the WIL2-S line were analyzed, one clone was selected for further cultivation and 350 of its subclones were analyzed by STR-PCR [18]. The scheme of the experiment is shown in Figure 1.

### 2.3. STR Profiling

STR profiles for each clone were assessed by PCR with primers to 19 STR loci and amelogenin locus available in COrDIS Plus multiplex kit (Gordiz Ltd., Moscow, Russia). Following markers were studied: D1S1656 (locus 1q42), D2S441 (2p14), D3S1358 (3p21.31), D5S818 (5q23.2), D7S820 (7q21.11), D8S1179 (8q24.13), D10S1248 (10q26.3), D12S391 (12p13.2), D13S317 (13q31.1), D16S539 (16q24.1), D18S51 (18q21.33), D21S11 (21q21.1), D22S1045 (22q12.3), CSF1PO (5q33.1), FGA (4q31.3), SE33 (6q14), TH01 (11p15.5), TPOX (2p25.3), VWA (12p13.31), amelogenin X (Xp22.1−22.3), and amelogenin Y (Yp11.2). For fragment analysis of PCR products, ABI 3130 genetic analyzer (Thermo Fisher Scientific, Waltham, MA, USA) was used. STR profiles were then analyzed using GeneMapper Software (v. 4-0).

### 2.4. Statistical Analysis

To test hypotheses about differences in distributions of categorical features in the comparison groups, the analysis of contingency tables was used. To assess the significance, a two-tailed Fisher’s test was used. As a measure of association for tables 2 × 2, the odds ratio (OR) is given with the corresponding 95% confidence interval (Ci). The target variable was the presence of LOH or EMAST at one or more of the studied loci.

## 3. Results

### 3.1. Cell Lines Verification by STR Profiling

Initially tested STR profiles of the cell lines taken in the study were in good correlation with that declared by ATCC (see Table 1). We tested other 11 STR loci additionally to that described by ATCC: D3S1358, D13S317, D21S11 are homozygous in the WIL2-S cell line, and D3S1358, D5S818, D16S539, vWA are homozygous in the Jurkat cell line (Table 1). These loci are only informative for the EMAST study in clones.

### 3.2. WIL2-S and Jurkat Primary Clones Analysis

Of the 18 primary clones of Jurkat, 7 have a loss of a Y-chromosome marker (the amelogenin locus); 5 have LOH on D12S391 (3 of these clones also have loss of the amelogenin Y marker); 1 clone—LOH on FGA (4q); 1 on CSF1PO (5q33.1); and one on D8S1179 (8q). EMAST was found in ten clones. Only four clones retained the STR profile of the original Jurkat sample (Figure 2). Of the 21 primary WIL2-S clones, 1 clone lost the amelogenin Y marker. It is clear that the proportion of aberrant clones is significantly higher in the Jurkat line (OR = 70.0 (Ci 95% 7.05–694.90) *p* < 0.0001).

### 3.3. WIL2-S and Jurkat Subclones Analysis

Upon subsequent cloning of selected primary clones, it was found that out of 350 subclones of the STR-stable WIL2-S clone, only two (0.57%) have lost the Y-chromosome amelogenin marker. One LOH at the D10S1248 locus (10q26.3) was found in only one of the 350 clones (0.29%). That means spontaneous somatic aberrations occur in the genome of non-tumor origin cell culture in less than 1 out of every 100 cultured cells (Appendix A). In the case of tumor cell line, LOH was observed in 26 out of 180 Jurkat-B4 subclones (14.4%). Of these, LOH in 5 markers was observed in two clones, and the patterns of aberrant markers in these two clones were identical (Jurkat-B4 115, Jurkat-B4 116, Appendix A). In two more clones, LOH was detected in two markers, aberrant clones did not match, and in 22 clones, LOH was found in only one STR locus. Thirty-five of 381 Jurkat-O1 subclones (9.1%) had LOH for one marker. EMAST was detected in 53 of 180 (29%) Jurkat-B4, of which 43 clones differed in one marker, 9 clones differed from the original culture in two markers, and one clone had EMAST in three STR markers (Appendix A). In Jurkat-O1 subclones, EMAST was observed in 111 of 381 subclones (29%), respectively. In 92 clones, EMAST was detected for one marker, in 12 clones for two markers, in one clone four mutant loci were observed, in five clones EMAST was detected in five loci, and in three of them the patterns of aberrant loci were identical (Jurkat-O1 379, Jurkat-O1 380, Jurkat-O1 381, Appendix A). In two other clones an absolutely identical aberrant STR profile was observed, which did not coincide with that of the three clones described above (Jurkat-O1 375, Jurkat-O1 378, Appendix A). Only one clone had six loci with EMAST. It should be noted that if EMAST manifests itself randomly for different STR markers.

The frequency of LOH occurrence in subclones for certain STR loci differs, and, apparently, is a pattern specific for this particular acute T-cell leukemia cell line (Figure 3 and Figure 4).

To understand whether the frequency of occurrence of aberrant clones differs between the offspring of a stable genetic profile of Jurkat-B4 and an aberrant profile of Jurkat-O1, we analyzed the contingency tables. It should be noted that according to this parameter, the clones are initially in unequal conditions. Loss Y and LOH D12S391 identified in Jurkat-O1 also appear in Jurkat-B4 subclones. Therefore, we decided to carry out the two-step analysis. At the first step, chromosomal lesions were taken into account in all the studied loci (OR = 1.67 (Ci 95% 0.97–2.86) *p* = 0.0618).

In the second step, only those lesions absent in the parental clone were analyzed. When the D12S391 and AmeloY loci were excluded from the comparative analysis, the accumulation of aberrations for all other loci was almost identical in the progeny samples of the two clones (*p* = 0.9216). The next task was to investigate the relationship between the accumulation of mutations in cells and the growth rate of clones. Since the selection of grown subclones was carried out every 2–3 days for a month and the clones were given continuous numbering, it seems correct to conduct a comparative analysis of the occurrence of aberrations by dividing the clone samples into equal parts with earlier clones and later clones.

### 3.4. Comparison of Aberrations Frequencies in Fast-Growing, Intermediate, and Slow-Growing Jurkat Subclones

The Jurkat-O1 sample was divided into three equal parts, in the first subclones 1–127, in the second subclones 128–254, and in the third subclones 255–381. The frequencies of LOH events (7/127 versus 10/127 versus 18/127) and EMAST events (41/127 versus 42/127 versus 68/127) for the three groups were analyzed in pairs in contingency tables using the Chi-square test. The samples of rapidly growing and normally growing clones practically did not differ from each other in terms of accumulation of events; therefore, they were combined into one group. The frequency of LOH in Jurkat1-O1 subclones was significantly higher in clones with a low growth rate. In slow-growing clones, LOH was found in 18 (14.17%) of 127 clones; in normally growing clones LOH was found in 17 (6, 69%) of 254 clones (OR = 2.30 (Ci 95% 1.14–4.64) *p* = 0.0172).

At the same time, the frequency of detection of clones with EMAST in Jurkat-O1 subclones was comparable in groups with slow and normal growth rates. In slow growing clones, loss of heterozygosity was found in 38 (29.92%) of 127 clones; in normally growing clones, loss of heterozygosity was found in 73 (28.74%) of 254 clones (*p* = 0.8110). But the mutation burden in the population of slow growing clones was significantly higher due to the presence of clones with multiple EMAST aberrations (in the group of slow growing clones, the proportion of clones with aberrations over 3 was 6/127 (4.72%), in the group with a normal growth rate—1/254 (0.39%) (OR = 12.56 (Ci 95% 1.49–105.36) *p* = 0.0030) (Figure 5).

Sample of Jurkat-B4 subclones was also divided into three parts. In the first group subclones 1–60, in the second group subclones 61–120, in the third group subclones 121–180 were included. And similarly, the first and second groups were combined into one sample due to the coincidence of the frequencies of event accumulation. The frequency of LOH in Jurkat -B4 subclones was also significantly higher in clones with a low growth rate. In slow growing clones, LOH was found in 13 (21.67%) of 60 clones; in normally growing clones LOH was found in 13 (10.83%) of 120 clones (OR = 2.28 (Ci 95% 0.98–5.29) *p* = 0.0513). The frequency of clones with EMAST in Jurkat -B4 subclones was significantly higher in clones with a low growth rate; 24 (40.00%) of 60 clones vs 30 (25.00%) of 120 clones (OR = 2.00 (Ci 95% 1.03–3.88) *p* = 0.0384) (Figure 6).

Only 60% of the Jurkat-B4 subclones and 64% of the Jurkat-O1 subclones retained an STR profile that fully matched that of the initial cell culture. That means new aberrations appear more often than in every third cell of a clone culture of tumor origin. We found 31 unique aberrant STR profiles with an incidence in the range of 0.6–6% in Jurkat-B4 subclones and 45 unique STR profiles with an incidence in the range of 0.3–3.4% in Jurkat-O1 subclones. Data on the distribution of clones with aberrant profiles relative to their growth rate are presented in Appendix A.

## 4. Discussion

Our simple in vitro model of clonal evolution fully reflects all the data previously described by many authors and explains the difficulties in determining the tumor heterogeneity. Back in 1976, Peter C. Nowell suggested that most neoplasms originate from a single parent cell, and tumor progression is the result of acquired genetic variability of the parent clone, which allows sequential selection of more aggressive subclones [19]. Populations of tumor cells appear to be more genetically unstable than normal cells. Genetic instability and subsequent selection of clones led to the fact that advanced human malignancies acquire distinct karyotype and biological features. Therefore, certain tumors may require specific therapy, that may be impaired by the emergence of therapy-resistant subclones. Understanding and controlling the tumor evolution before it reaches the advanced stage is a very important task in clinical oncology. In 1977, Fidler et al. [20] experimentally demonstrated differences in mouse melanoma tumor cells. Clones obtained in vitro from the original cell culture of murine malignant melanoma differed significantly in their ability to form metastatic colonies in the lungs when intravenously inoculated into syngeneic mice. The authors explained this result by the fact that the initial tumor is heterogeneous and that highly metastatic variants of tumor cells already exist in the parental population. In the past decade, with the development of new technologies, more and more complex and sensitive methods have been used to study tumor heterogeneity. However, the simple approach of splitting tumor biopsies into small fragments in order to capture minor clones is still being successfully applied. For example, Gerlinger et al. performed exome sequencing, chromosome aberration analysis, and ploidy profiling on several spatially separated samples derived from primary renal carcinomas and associated metastatic sites. Signs of gene expression of good and poor prognosis were found in different areas of the same tumor. Analysis of allelic composition and ploidy profiling revealed extensive tumor heterogeneity [21].

Creso et al. demonstrated in a mouse model that the transplantation of cancer cells at doses of clonal cells to multiple recipients allows separation and investigation of subclones. Following engraftment of human cancer cells into immunocompromised mice, the composition of a particular subclone was examined by prospective cell purification. Sequential cancer cell transplants have made it possible to track further clonal evolution. In the same work, the authors demonstrated the transformation of genetic characteristics and natural properties of tumor clones, the so-called chemo-induced clonal evolution, caused by oxaliplatin, which was systematically administered to mice after tumor formation [22].

Chemo-induced clonal evolution was also noted in hematological malignancies. Ding et al. in 8 AML patients confirmed hundreds of somatic mutations in tumor cells obtained in relapse compared to primary tumor cells using deep sequencing to determine the mutation spectrum. In addition to the discovery of new, intermittently mutated genes (e.g., WAC, SMC3, DIS3, DDX41, and DAXX) in AML, two major patterns of clonal evolution during AML relapse were also found: (1) the founder clone in the primary tumor acquired mutations and evolved into a clone of relapse, or (2) a minor subclone of the founder clone survived the initial therapy, acquired additional mutations, and proliferated at relapse. These data show that AML recurrence is associated with the addition of new mutations and clonal evolution, which partially could result from the chemotherapy patients receive to establish and maintain remissions [23].

By analyzing the paired samples of multiple myeloma patients before treatment and at the time of relapse by comparative genomic hybridization, Keats et al. found that every third patient had DNA copy number changes associated with clonal heterogeneity at diagnosis. Another third of patients showed the appearance of new aberrations in addition to those identified at the onset of the disease, which is consistent with the hypothesis of linear evolution. These groups included virtually all high-risk patients, suggesting that high-risk tumors are less stable and more likely to change over time [24].

Very similar findings were made by Jiang et al. exploring DLBCL by sequencing rearranged VDJ junctions in 14 paired tumor samples at onset and relapse, among which 7 pairs were further characterized by exome sequencing. The authors also note two distinctive modes of DLBCL clonal evolution at relapse: an early divergent mode with tumor clones, responsible for relapse being present at the diagnosis, and a late divergent type, in which recurrent tumors develop directly from primary tumor clone [25].

Oshima et al. studied paired tumor samples at the onset and recurrence of childhood ALL using whole exome and whole genome sequencing. Numerous relapse-associated mutations associated with resistance to chemotherapy have been identified, in particular mutations that activate the RAS-MAPK pathway. In some cases, retention or emergence of RAS mutant clones upon relapse was noted, while in others RAS-mutated clones present at diagnosis have been replaced by wild-type RAS populations. Therefore, an impact for both positive and negative selection on the clonal evolution of RAS-mutated leukemia is observed [26]. Smirnova et al. assessed the stability of IG and TCR gene rearrangements in ALL in adults. Five out of 6 (83%) of the studied patients had differences in clonal rearrangements at onset and relapse, which indicates the instability of clones during polychemotherapy, i.e., a probable chemo-induced clonal evolution [27]. Malcikova et al. showed that for CLL, the minor TP53-mutated clones with a low allelic load detected by NGS, in most cases became dominant at the first or subsequent relapse [28].

NGS also provides information about subclones of solid tumors and is used to study tumor metastatic patterns. Intratumor heterogeneity (ITH) may become a new clinical prognostic indicator. Yu et al. conducted a meta-analysis to investigate whether ITH can serve as a valuable prognostic indicator in solid tumors. The analysis included studies from the PubMed, Embase, Cochrane, and Web of Science databases as of 10 October 2020. ITH-based studies with prognostic information available were included. A total of 9804 patients with solid tumors from 21 studies were included. It has been shown that a high level of ITH is associated with a shorter overall, event-free, and disease-free survival in general [29].

Su et al. studied transcriptome landscape of childhood acute megakaryoblastic leukemia (AMKL), an extremely unfavorable variant of AML, using the scRNA-seq (single-cell RNA sequencing) technique. ITH in AMKL was found in different tumor markers and different patterns of DNA copy number variations (CNV) [30].

Unfortunately, it is difficult to assess the ITH by routine methods before starting therapy. However, this information could be extremely valuable for decision-making concerning therapy options. We have shown in in vitro model that ITH potential can be very high—up to a third of tumor clone progeny cells may acquire new aberrations. Moreover, it is important that both a clone that already has certain genetic lesions and a clone without identified tumor markers can evolve. Not all the newly acquired mutations may relate to the emergence of resistance or metastasis. However, the wider the range of mutations, the more likely it is that the key ones determining the aggressive nature of the tumor will appear. The proposed in vitro model of tumor evolution also demonstrated that the accumulation of mutations occur permanently during cell culture. We have shown that the offspring of one cell can give rise to a number of different aberrant clones. Due to the wide range of mutations, all of them are characterized by a low allele frequency, below 10% (Figure 3 and Figure 4), and affect an average of 1–2% of the cell population, that complicates their identification in routine practice. However, the expansion of minor clones during therapy, can be a critical event. Here we report the association of increased aberrational load with the slowly growing subclones. Perhaps this is a feature of this cell line, but it is possible that slow-growing clones could be resistant to the therapy due to their greater genetic plasticity. And their late detection explains the change in the mutational landscape in the tumor cell population from diagnosis to relapse.

## 5. Conclusions

According to our model, spontaneous aberrations in the genome of cells of non-tumor origin occur in less than 1 of 100 cultured cells. While in the cells of tumor origin, new aberrations appear in 1 or even more of 3 cultured cells. At the same time, a statistically significant relationship was found between the accumulation of aberrations in the pool of subclones and the low rate of cell growth. STR analysis seems to be a convenient tool for the routine study of tumor heterogeneity and assessment of its evolutionary potential. One can speculate that this approach could be applied for the analysis of primary patient tumor cell culture to obtain information concerning the evolutionary potential of the tumor cells that may be useful for the therapy selection.

## Figures and Tables

**Figure 1 genes-14-00571-f001:**
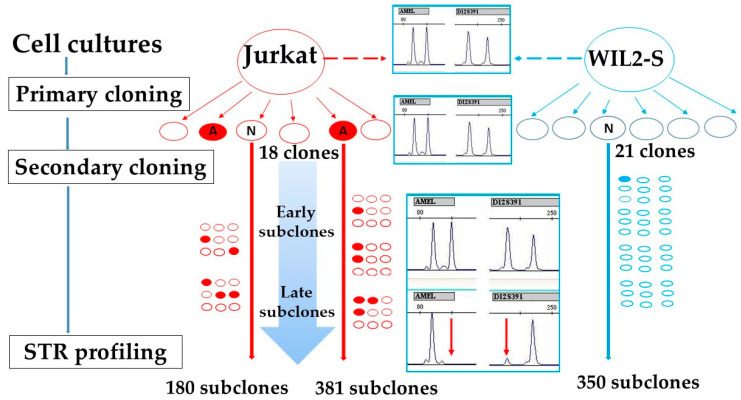
Design of the experiment. The Jurkat cell line, its clones, and subclones are marked in red, the WIL2-S cell line, its clones, and subclones are marked in blue. A—aberrant STR profile, N—normal STR profile. The large blue arrow with gradient coloring symbolizes the increase in the number of aberrations in the late subclones.

**Figure 2 genes-14-00571-f002:**
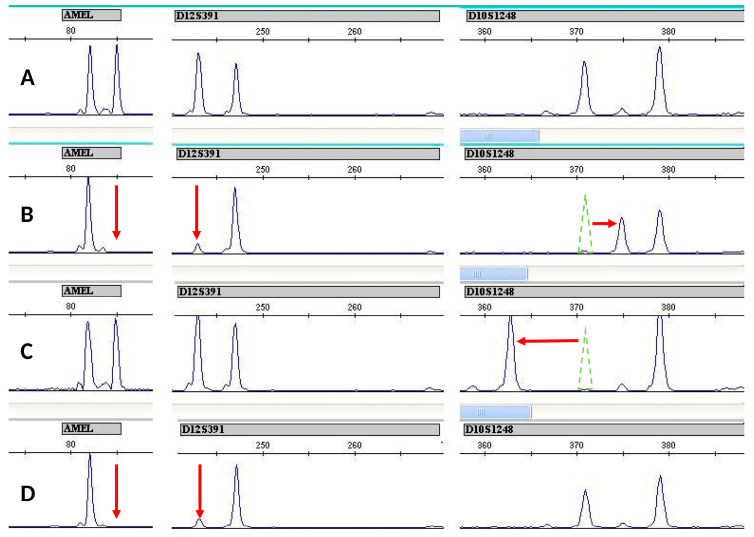
Variants of aberrations observed in STR profiles of primary Jurkat clones. (**A**) Stable clone Jurkat-B4 (author’s numbering), (**B**) clone with the loss of the Y-chromosome marker, LOH D12S391, EMAST +4 nucleotides at allele 14 D10S1248, (**C**) EMAST −8 nucleotides at allele 14 D10S1248, (**D**) Jurkat-O1 (author’s numbering) clone with the loss of the Y-chromosome marker, LOH D12S391. Clones A and D were selected for further cloning.

**Figure 3 genes-14-00571-f003:**
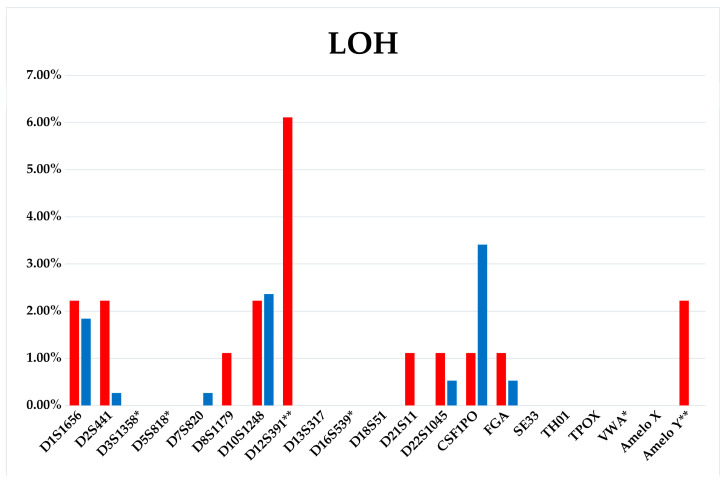
LOH distribution diagram among the STR loci of the Jurkat-B4 (red bars) and Jurkat-O1 (blue bars) subclones. *—loci homozygous in the control—Jurkat culture, **—loci aberrant in the primary clone Jurkat-O1.

**Figure 4 genes-14-00571-f004:**
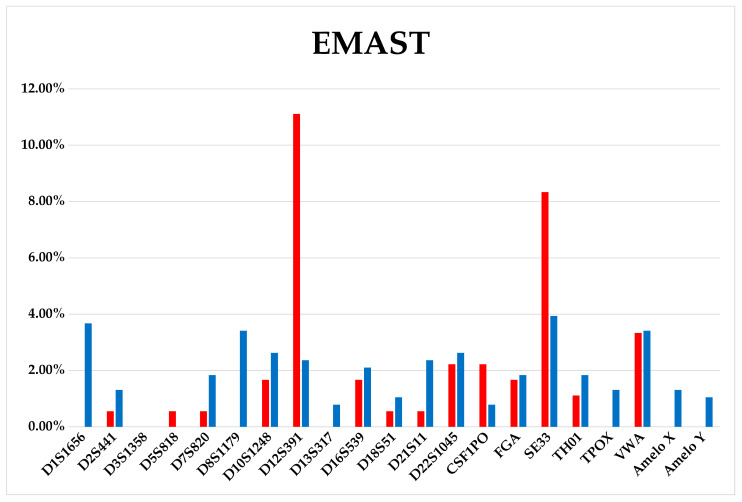
EMAST distribution diagram among the STR loci of the Jurkat-B4 (red bars) and Jurkat-O1 (blue bars) subclones.

**Figure 5 genes-14-00571-f005:**
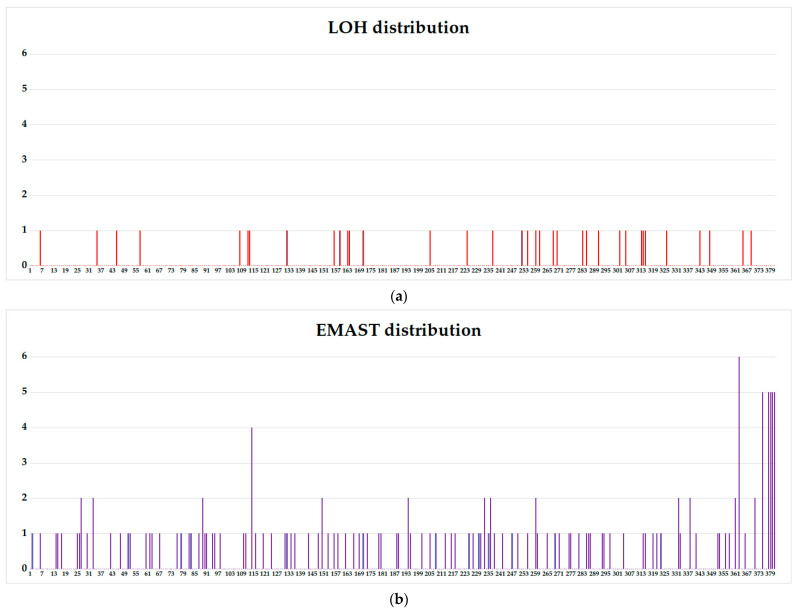
Distribution diagram of LOH and EMAST events between Jurkat-O1 subclones. In the x-axis are the numbers of subclones, from the earliest to the latest, slowly growing. In the Y axis—the number of aberrations in one clone (no more than 6). (**a**) Distribution of LOH; (**b**) distribution of EMAST.

**Figure 6 genes-14-00571-f006:**
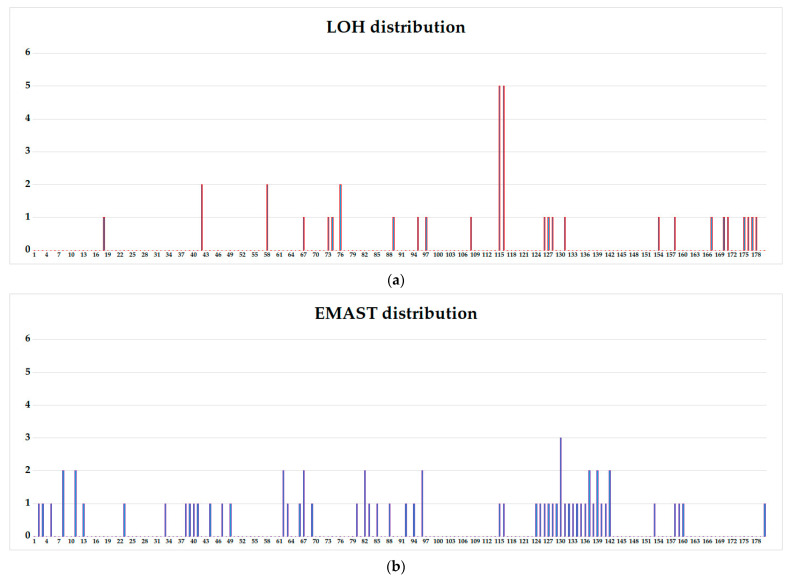
Distribution diagram of LOH and EMAST events between Jurkat-B4 subclones. In the x-axis are the serial numbers of clones, from the earliest to the latest, slowly growing. On the Y axis—the number of aberrations in one clone (no more than 6). (**a**) Distribution of LOH; (**b**) distribution of EMAST.

**Table 1 genes-14-00571-t001:** STR profiling of cell lines.

STR Locus	WIL2-S Alleles ^1^	WIL2-S Alleles ^3^	Jurkat Alleles ^2^	Jurkat Alleles ^3^
Amelogenin	X,Y	X,Y	X,Y	X,Y
CSF1PO	11,12	11,12	11,12	11,12
D13S317	11	11	8,12	8,12
D16S539	11,12	11,12	11	11
D5S818	12,13	12,13	9	9
D7S820	9,11	9,11	8,12	8,12
THO1	8,9.3	8,9.3	6,9.3	6,9.3
THPOX	8,11	8,11	8,10	8,10
vWA	17,20	17,20	18	18
D1S1656	-	14,15.3	-	15.3,16.3
D2S441	-	12,15	-	15,16
D3S1358	-	16	-	15
D8S1179	-	10,13	-	13,14
D10S1248	-	14,16	-	14,16
D12S391	-	17,22	-	22,23
D18S51	-	11,16	-	12.2,21
D21S11	-	28	-	30.2,32.2
D22S1045	-	15,16	-	14,17
FGA	-	20,22	-	20.2,21.2
SE33	-	16,18	-	16,18

^1^ According to ATCC CRL8885, ^2^ according to ATCC TIB152, ^3^ our own data.

## Data Availability

Data available from corresponding author upon reasonable request.

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
