# Peer review of "Accumulation of STR-Loci Aberrations in Subclones of Jurkat Cell Line as a Model of Tumor Clonal Evolution"

_genes, 2023, doi:10.3390/genes14030571_

Round 1
Reviewer 1 Report
In the paper entitled “Accumulation of STR-loci aberrations in subclones of Jurkat cell line as a model of tumor clonal evolution” by Natalya Risinskaya and colleagues, authors have examined the STR loci in tumorous JURKAT, and non-tumorous WIL2-S cell line, and, additionally, performed limited dilution cloning in order to follow acquisition of additional genomic lesions and evolution of selected clones. Overall the paper is difficult to read, mostly due to the lack of the clear explanation of why authors decided to perform certain experiments (e.g., cloning specific clones), lack of the clear separation of different subheadings of the topics that authors tried to tackle in the result section, as well as uninformative figures which do not really contribute to the overall quality of the manuscript. For example, presenting phylogenic tree or a fish plot from individual cultured clone is more informative than bar plots. Furthermore, discussion is rather long and authors mostly give an overview of the previous findings on the topic of the clonal evolution in cancer, with limited information about ALL and lack of relevant literature on this topic. For example, PMID: 25790293, 32793890, 33540666, 33147938, 25985233, 34597466, 32086311, 29365312, 31852987, and etc. Authors discuss their own work only in the last paragraph of the discussion and mostly summarize their findings, instead of discussing them in the context of the current literature. English language used is overall fine, but can benefit from further editing. Finally, authors should make clear what is the benefit of their study, compared to other published work, dealing with the topic of clonal evolution using more sensitive and high-throughput techniques. In general, the study lacks novelty and many of the observations, including higher mutational burden in fast proliferating clones, were known from previous studies.
Author Response
Dear Reviewer,
We are really grateful for the extremely helpful comments and criticism. We did our best to improve our manuscript according to your suggestions.
"Overall the paper is difficult to read, mostly due to the lack of the clear explanation of why authors decided to perform certain experiments (e.g., cloning specific clones)", Please find in text (lines 44-49) our explanation of object and design choice with the referrences to suggested literature which are also added [11, 12].
"lack of the clear separation of different subheadings of the topics that authors tried to tackle in the result section" We added subheadings for Materials and Methods and Results sections.
"For example, presenting phylogenic tree or a fish plot from individual cultured clone is more informative than bar plots". Unfortunately, the phylogenic tree or a fish plot from individual cultured clone, that might be more informative in case with fewer subclones, would be overloaded in our case with 31 and 45 unique variants of aberrant STR-profiles for two individual clones. We added two tables to the Supplementary materials with the data on the distribution of subclones with aberrant profiles relative to their growth rate.
"English language used is overall fine, but can benefit from further editing". The manuscript was evaluated and improved by a native speaker.
Sincerely, Natalya Risinskay
Reviewer 2 Report
In this manuscript Risinskaya et. al. report an in vitro model to study tumor clonal evolution using STR profiling. Following STR profile of subclones of Jurkat, a tumor cell line, and a non-tumor cell line, WIL2-S, they have shown significantly higher diversion of clonal STR profiles in Jurkat cell line compared to WIL2-S. A temporal dynamics of STR changes were monitored for the clonal evolution in culture. Authors suggest that increased genomic aberration in clonal line of Jurkat is not merely due to higher cell proliferation since they did not find a correlation between cell proliferation rate with increased genomic aberration. Despite current development of single cell genomics, this model of tumor evolution study has potential of broader interest due to is more accessibility. Overall, the manuscript and the figures are presented coherently, however, the Result and Materials and Methos sections needs to be formatted for the clarity of the manuscript.
1. Inclusion of subheadings for Materials and Methos section would be helpful for readers of the article. For example, Cell culture and treatment, DNA isolation and PCR, and Statistical analysis etc.
2. Similarly break the Results section into coherent subsections.
3. Define all abbreviation at their first use in the manuscript in addition to keywords provided in the footnote.
Author Response
Dear Reviewer,
We appreciate your valuable comments and positive estimation of our work.
We added subheadings as you suggested for Materials and Methods, and Results sections. Furthermore, we also defined abbreviations with the first appearance in the manuscript.
Sincerely, Natalya Risinskaya
Round 2
Reviewer 1 Report
Authors answered most of my comments and corrected manuscript accordingly.